# Smoking Cessation Messages for Pregnant Aboriginal and Torres Strait Islander Women: A Rapid Review of Peer-Reviewed Literature and Assessment of Research Translation of Media Content

**DOI:** 10.3390/ijerph18179341

**Published:** 2021-09-04

**Authors:** Tara Flemington, Gina La Hera-Fuentes, Michelle Bovill, Allison Hart, Jessica Bennett, Nicole M. Ryan, Gillian Sandra Gould

**Affiliations:** 1Susan Wakil School of Nursing and Midwifery, University of Sydney, Sydney, NSW 2006, Australia; tara.flemington@health.nsw.gov.au; 2School of Medicine and Public Health, The University of Newcastle, Callaghan, NSW 2308, Australia; gina.laherafuentes@newcastle.edu.au (G.L.H.-F.); michelle.bovill@newcastle.edu.au (M.B.); allison.hart@newcastle.edu.au (A.H.); jessica.bennett@newcastle.edu.au (J.B.); nicole.ryan@newcastle.edu.au (N.M.R.); 3Faculty of Health, Southern Cross University, Coffs Harbour, NSW 2480, Australia

**Keywords:** pregnant women, smoking cessation, Aboriginal and Torres Strait Islander health, social media

## Abstract

This review summarized literature about knowledge, attitudes, and beliefs of Aboriginal and Torres Strait Islander women from Australia who smoke during pregnancy, then examined the extent that existing health promotion materials and media messages aligned with evidence on smoking cessation for pregnant Aboriginal and Torres Strait Islander women. Knowledge, attitudes, and beliefs of pregnant Aboriginal women who smoke tobacco were identified in the literature. Health promotion campaigns were retrieved from a grey literature search with keywords and social and professional networks. Key themes from peer-reviewed papers were compared against the content of health promotion campaigns using the Aboriginal Social and Emotional Wellbeing Model, the Behavior Change Wheel and thematic analysis. Eleven empirical studies and 17 campaigns were included. Empirical studies highlighted women sought holistic care that incorporated nicotine replacement therapy, engaged with their family and community and the potential for education about smoking cessation to empower a woman. Health promotion campaigns had a strong focus on ‘engagement with family and community’, ‘knowledge of risks of smoking,’ ‘giving up vs cutting down’ and ‘culture in language and arts’. There were similarities and variances in the key themes in the research evidence and promotion materials. Topics highly aligned included risks from smoking and quitting related issues.

## 1. Introduction

“*… we move towards forging a positive way forward for those coming behind us, by creating a new discourse on Aboriginal women, one filled with messages of strength and hope for younger women and for our people*.” Gregory [1].

Inequities in health policy and service provision manifest in a variety of ways, including disproportionality in high-risk behaviors such as tobacco smoking. The high prevalence of smoking amongst Indigenous peoples globally is highlighted in the preamble to the World Health Organization’s Framework Convention on Tobacco Control [2].

In Australia, Aboriginal and Torres Strait Islander women have higher smoking rates compared with non-Aboriginal women, observing a percentage of 44% compared with 11% [3]. Therefore, promoting a smoke-free pregnancy benefits Aboriginal and Torres Strait Islander women and their babies, including reducing the risk of perinatal death, preterm birth, small for gestational age, and severe neonatal morbidity [2]. While Aboriginal and Torres Strait Islander people are more likely to attempt to quit smoking than other Australians, they have less success [4]; specifically, during pregnancy, Aboriginal and Torres Strait Islander women are highly motivated to quit smoking, but they are widely unsupported to do so by health providers [5].

The success of government strategies that prioritize smoking cessation is dependent on adopting an evidence-based approach that is tailored to the specific contexts of pregnant Aboriginal and Torres Strait Islander women. However, there is little research how evidence regarding the best ways to support smoking cessation and inform the health messages used to promote smoking cessation among Aboriginal and Torres Strait Islander pregnant women. While Indigenous peoples comprise diverse groups with a variety of strengths and cultures, the present-day impacts of colonization are markedly similar. For Aboriginal and Torres Strait Islander populations, smoking is tightly linked to the nation’s history of colonization, and subsequent government-sanctioned policies of racism, discrimination, and oppression. Settlers cultivated tobacco dependency as a ‘civilizing’ or ‘taming’ influence [6], and Aboriginal and Torres Strait Islander peoples often worked in brutal circumstances in exchange for tobacco. Accessibility to tobacco played a part in the migration of Aboriginal and Torres Strait Islander communities to areas of white settlement, contributing to the severance of connections with lands, languages, histories and cultures [6].

In spite of these severed connections, Aboriginal and Torres Strait Islander communities, culture, and women remain strong. For Aboriginal and Torres Strait Islander women, theirs is a story of ‘trauma and triumph’ [1]. In concert with individual protective factors such as inner strength, maintaining a strong sense of identity and practicing culture, this resilience is also attributable to the strong connections and relationships between Aboriginal and Torres Strait Islander women in friendship groups, family and community [1]. Smoking is often a part of these social interactions, and for pregnant Aboriginal and Torres Strait Islander women in a collectivist kinship system, this can provide a sense of belonging and connection [7,8]. It follows then that social media campaigns targeting smoking cessation for pregnant Aboriginal and Torres Strait Islander women would need to draw on these key aspects of women’s lives and their culture.

Improved access to internet communication technologies and exponential growth in smartphone usage locates social media as an ideal complement to traditional public health smoking cessation media campaigns. Social media (for example, Facebook, Twitter, Instagram, and WhatsApp) is potentially a powerful channel for health promotion message delivery. However, the uptake, usage and style of interaction on social media platforms vary greatly by population groups. For Aboriginal and Torres Strait Islander Australians, social media platforms have potential to ‘level the playing field’ somewhat, dismantling existing hierarchical power structures and returning ownership of Aboriginal and Torres Strait Islander culture and voices to Aboriginal and Torres Strait Islander communities. They can be a place to affirm Indigenous identities, establish a sense of power and control over the portrayal of online identity, share and grow traditional culture, strengthen community and family connections, and participate in health promotion programs [9].

In mainstream populations, social media smoking cessation interventions have strong feasibility and acceptability [10] and some success in the initiation and maintenance of smoking cessation [11]. However, the overall quality of smoking cessation mobile applications and their adherence to the evidence base and smoking cessation guidelines is variable [12,13]. While there is limited research exploring the effectiveness of culturally targeting media messages for smoking cessation, early campaigns have been effective in terms of changes in knowledge, attitude and behavior for Indigenous peoples globally [14]. In a national two-wave survey of 739 Aboriginal and Torres Strait Islander respondents who smoke, those who recalled any local advertising were significantly more likely to have attempted to quit smoking (58% vs. 39%, AOR: 2.03) [15]. The tobacco smoking campaign for pregnancy, Quit for You, Quit for Two, had prompted the recall of 61% in women who had been pregnant or contemplating pregnancy, 75% of currently smoking respondents intended to take further action and 40% considered quitting [16].

A series of recent reviews have explored various aspects of the journeys of pregnant Aboriginal and Torres Strait Islander women and smoking cessation, including experiences, perspectives and values [7]; barriers and facilitators for smoking cessation [17]; the upholding of empowerment during smoking cessation interventions [18]; and knowledge, views, and barriers to cessation [19]. To date, no systematic review has specifically focused on best evidence approaches for social media messages for smoking cessation among pregnant Aboriginal and Torres Strait Islander women or pregnant Indigenous women globally.

The aims of this rapid review are twofold:*Part 1.* Summarize the literature about knowledge, attitudes, beliefs, and values of Aboriginal and Torres Strait Islander women who smoke during pregnancy using the Aboriginal Social and Emotional Wellbeing Model. Followed by using the Behavior Change Wheel (BCW) and the Capability, Opportunity, Motivation-Behavior (COM-B) model to identify where best-practice approaches were used to develop health promotion messaging for health promotion campaigns. The summary will focus on information to inform social media messages; and*Part 2.* Evaluate the extent to which existing smoking cessation messages for pregnant Aboriginal and Torres Strait Islander women in Australia reflect this evidence base and make recommendations to improve smoking cessation messages for Aboriginal and Torres Strait Islander women.

## 2. Materials

Rapid reviews adopt a modified approach to the systematic review process to establish an overview of pertinent evidence in a relatively shorter space of time, that should be less than 6 months; and exclude some of the steps, for example not using two reviewers. In the absence of an accepted protocol, our approach was informed by Haby et al. [20]. They identified that a common feature of rapid reviews is the close relationship between the reviewers and the end-user. In our case, this review will inform a social media campaign to be funded by the Australian Department of Health. The review is registered with PROSPERO (CRD42020169852).

### 2.1. Search Strategies and Selection of Materials

Two search strategies were adopted. To address the first aim, part 1, review evidence in the literature to establish the knowledge, beliefs and values of pregnant Aboriginal and Torres Strait Islander women who smoke during pregnancy, a traditional keyword search of databases with inclusion and exclusion criteria was conducted in July 2019 as presented in Table 1.

Next, to address the second aim, part 2, and evaluate the extent to which smoking cessation messages for pregnant Aboriginal and Torres Strait Islander women in Australia reflect the evidence base, health promotion materials for this audience available in the public domain were collated and evaluated by hand. In addition to the online approach presented in Table 1, these health promotion materials were sourced through a snowball strategy that included a call-out through social and professional networks, for example, the Australian Department of Health Tackling Indigenous Smoking program and regional projects, Cancer Councils, and Aboriginal Community Controlled Health Services. Searches, screening, and analysis were conducted by Aboriginal and non-Aboriginal members of the research team.

Two Aboriginal researchers (JB and MB) did first search strategy, JB screened the articles and TF (non-Aboriginal researcher) conducted the second screening. MB provided leadership with GG (non-Aboriginal researcher) overseeing the methodology, formal analysis and writing and editing the review.

Ethical considerations were taken into account. This study was performed with Aboriginal and non-Aboriginal researchers with experience in research focused in Aboriginal and Torres Strait Islander population. This review considers the consumer’s perspective, Aboriginal and Torres Strait Islander people when identifying the salient messages and the health care perspective when identifying the campaigns or the response of the system.

Since 2008, the Australian Federal Government has been targeting smoking as a key component to improve the life expectancy for Aboriginal and Torres Strait Islander peoples [21], which includes pregnant women. This significant milestone in 2008 was used to inform the initial search date for identifying smoking related health campaigns and messaging for pregnant Aboriginal and Torres Strait Islander women.

### 2.2. Data Synthesis and Analysis

#### 2.2.1. Part 1

Summary characteristics of the included articles were extracted and tabulated by TF, using headings related to author and date, aims, methods (setting, design, sample) and outcomes. Study findings were categorized in accordance with relevant domains of the BCW, COM-B and Aboriginal Social and Emotional Wellbeing Model frameworks. An integrated design approach was adopted for synthesis of qualitative and quantitative findings. In this approach, because the studies address similar research questions, the impacts of methodological differences are minimized and the findings of both approaches are readily synthesized for meaning extraction (Heyvaert et al. (2016) as cited in [22]). In respect of complexity of concepts of health and well-being and the diversity and strengths of Aboriginal and Torres Strait Islander peoples, our first preference was to synthesize data following a culturally specific health promotion behavior change framework. In the absence of such a model, we have adopted a bicultural approach.

First, the literature and the health promotion materials were classified following the seven domains of the Aboriginal and Torres Strait Islander Social and Emotional Wellbeing Model, comprising the connection of self to body; mind and emotions; family and kinship; community; country; and spirit, spirituality, and ancestors [23]. In this context, the self is viewed as inseparable from and embedded within, family and community [23].

The Aboriginal and Torres Strait Islander Social and Emotional Wellbeing Model was used as it proposes a multidimensional view of health from an Aboriginal and Torres Strait Islander standpoint. It was a framework that was developed to help clinicians address the uncertainty in their work about how to approach Aboriginal and Torres Strait Islander Health [23]. Consequently, the model clarifies guiding principles and broad areas of wellbeing that need to be considered when working in this field with Aboriginal and Torres Strait Islander peoples. While not intended by the originators as a definitive model or way to classify literature or health promotion programs, our exploratory use for this purpose, we believe, is warranted. The model was used pragmatically to help explore the focus and scope of messages related to an Aboriginal and Torres Strait Islander holistic concept of health, rather than a model to assess a program or intervention.

Secondly, we evaluated the extent to which the literature and the health promotion materials aligned with the Behavior Change Wheel, an accepted theoretical framework for behavior change. The Behavior Change Wheel has been used previously to examine factors affecting smoking cessation for pregnant Aboriginal and Torres Strait Islander women [24]. The Behavior Change Wheel is a parsimonious model that has the COM-B system made up of six sources of behavior at its inner hub, a middle wheel layer denoting nine intervention functions and an outer layer of seven policy categories [25]. The COM-B constructs comprise psychological and physical capability; physical and social opportunity; and reflective and automatic motivation [25]- all of which have an impact on behavior. The intervention functions are the variety of approaches that can be utilized to encourage health behavior change and include modelling, education, and incentivization. Policy categories, which include fiscal measures and legislation, are beyond the scope of the materials examined in this review.

#### 2.2.2. Part 2

The health promotion materials were tabulated and categorized according to the campaign name, a summary of message content, target audience, call to action, and frame. The framework analysis included whether the message was a loss versus gain frame (for example, ‘you will lose money if you continue smoking’ vs. ‘you will save money if you quit smoking’); the inclusion of an appeal that is efficacy-based or threat (‘you too can quit smoking’ vs. ‘if you smoke your baby will be born sickly’), and an infant-centric or maternal-centric message (‘healthier baby’ vs. ‘more energy for you’).

Following this summary classification of data for part 1 and part 2, a thematic analysis of the peer-reviewed literature was planned and undertaken by a non-Aboriginal researcher (TF). An inductive approach to analysis was adopted, with TF conducting coding and initial theme generation. This was then reviewed with an Aboriginal researcher (AH), by assessing the themes for alignment with the literature and Aboriginal cultural meanings. In this step, Aboriginal researcher (AH) utilized cultural knowledges and interpretations of the literature to the initial themes, applying an additional layer of understanding and meaning to the analysis. The themes were then refined and named in accordance with their eventual definitions. Health promotion materials were examined for the extent to which they aligned with the dominant themes from the research evidence.

## 3. Results

A total of 193 peer-reviewed articles were identified and screened by an Aboriginal researcher (JB). After removal of 166 articles for duplication and not meeting inclusion criteria, further screening was undertaken by TF, where 16 additional papers were excluded. Summary characteristics of the 11 included articles are presented in Figure 1 and Table 2.

All included articles except three were qualitative studies. The quantitative studies [27,28,29] were included because they had Aboriginal and/or Torres Strait Islander women as participants, and provided quantitative data to complement qualitative findings. In total, the 11 papers included 603 participants.

A total of 17 health promotion campaigns targeting smoking cessation for pregnant Aboriginal and Torres Strait Islander women were identified, and the characteristics of these campaigns are presented in Table 3. Most campaigns utilized a variety of media in their approach, including brochures, Film, posters, and social media. One campaign, *Stickin’ it up the Smokes* [30], had a dedicated social media presence. Other campaigns were located within existing smoking cessation or health promotion social media platforms, such as *Ready Mob* [31] and *Deadly Choices* [32]. Most of the campaigns (82%; 14/17) specifically targeted pregnant women, another two were aimed at all women, and only three included a broader audience such as the family. All calls to action were either explicitly to quit smoking, and/or referral to a health provider, e.g., local Aboriginal Medical Service, a specialist service such as the Quitline (*n* = 6) or a program such as Quit For New Life. Five campaigns used a loss frame, five used a gain frame, and 7 used both loss and gain. The majority (59%; 10/17) used an efficacy message, only two used a threat message and five used both efficacy and threat messages. About equal numbers used either a solely baby-centered message (*n* = 8) or both baby-centered and maternal-centered (*n* = 9) messages. No campaign only used maternal messages.

The alignment of the peer reviewed evidence and health promotion materials with the domains of the Aboriginal Social and Emotional Wellbeing Model and the Behavior Change Wheel Model are presented in Table 4 and Table 5. All included studies explored the domains’ Connection to Body’ (by exploring the act of smoking itself) and ‘Connection to Mind and Emotions.’ This was most often in the context of stress or stress management [27,28,29,33,34,36,38,40], although concepts of autonomy, aspirations and empowerment also featured [35,37,38]. The related domains of ‘Connection to Family and Kinship’ and ‘Connection to Community’ were considered in almost all studies. This was through peer support mechanisms [33,39] as well as the positive and negative impacts of family and community [28,29,34,36,37,38,39,40]. ‘Connection to Country and Culture’ were identified twice [35,38], and ‘Connection to Spirit, Spirituality and Ancestors’ was not addressed by any studies.

As with the research evidence, all health promotion materials were represented in the ‘Connection to Body’ domain for their smoking cessation topic area. However, unlike the research, health promotion materials were more likely to incorporate ‘Connection to Culture’ and ‘Connection to Mind and Emotions’ than any other domain of the Aboriginal Social and Emotional Wellbeing Model. ‘Connection to Culture’ was evident in a variety of ways that often overlapped with other domains. Language was a commonly used method to incorporate culture, whether by presenting an entire film in language for *Puyu paki (Don’t smoke—give it up)* [47], the bilingual campaign title for *Ngamari Free* [49], or incorporating Aboriginal ways of speaking in English in *Stronger Boorais* [41]. Traditional Aboriginal and Torres Strait Islander artwork was incorporated into most campaigns, and a hip-hop soundtrack by Aboriginal and Torres Strait Islander musicians was used to highlight a scene of self-efficacy and positive future aspirations in *Smoking: They Can’t Choose, It’s Up To You* [43]. ‘Connection to Country’ was identified twice [41,55], and ‘Connection to Spirit, Spirituality and Ancestors’ was identified once, in *Smoking: They Can’t Choose, It’s Up To You* [43].

All of the published studies and health promotion materials explored the Behavior Change Wheel /COM-B behavioral construct of psychological capability, with social and physical opportunity also featuring heavily. Education was the intervention function most frequently featured in both the research and health promotion materials. While health promotion materials also strongly featured intervention functions of persuasion and role modelling, the research studies were more likely to explore functions of enablement, followed by incentives and role modelling. *Stronger Boorais* and *Birthing in our Community* addressed all COM-B behavioral domains and were also well represented in the domains of the Aboriginal Social and Emotional Wellbeing Model [41,55].

From the included publications, a thematic analysis of the evidence base revealed three major themes of importance for pregnant Aboriginal and Torres Strait Islander women on their smoking cessation journey, provided below. These were divided into several subthemes.

**Theme** **1.***Holistic partnerships with women, their family and their community (subthemes: NRT, mechanism of support, impact of stress, and engagement with family and community)*.

Women sought holistic care that incorporated nicotine replacement therapy [28,33,34,39], engaged with their family and community [27,28,29,33,34,35,36,39,40], and addressed the relationship between stress and smoking cessation [27,28,29,39]. Women valued support mechanisms that were flexible, frequent, and also incorporated options for non-pharmacologic approaches [28,33,34,35,39].

**Theme** **2.***Empowerment through knowledge and roles models (subthemes: intrinsic beliefs and attitudes, knowledge of risks of smoking, giving up vs. cutting down, limitations of health provider practice, a woman’s autonomy and the role of family and community)*.

Whilst a woman’s intrinsic beliefs and attitudes towards smoking cessation were important [29,34,36,38,40]. The potential for education about smoking cessation to empower a woman and strengthen her autonomy in the quitting process was highlighted [28,29,35,36,39,40]. Furthermore, the limitations of health provider practices in this process reduced a woman’s chances of smoking cessation success, with health providers offering little or inconsistent advice, and having less respect in the community than a woman’s relatives and elders [34,36,37,38,40]. The importance of reflective motivation was seen in women who expressed regret for smoking decisions, nurturing hope for their own and their baby’s future, and the impact of marginalization and racism on these dreams [35,37,38].

**Theme** **3.***Culturally meaningful (subthemes: knowing the past, dreaming of the future, culture in language and arts, relationships with community, respect for relatives and Aboriginal and Torres Strait Islander communities)*.

The meaning and impact of Aboriginal and Torres Strait Islander culture for women who smoke was explored infrequently in the literature. When it was, women responded positively to materials and experiences that highlighted Aboriginal and Torres Strait Islander language and artistic design [35] and strengthened their connection to culture [33]. Aboriginal and Torres Strait Islander community and kinship relations were also important influences on an Aboriginal and Torres Strait Islander girl’s or woman’s initiation into smoking [34,37], and for some young girls it was seen as a way to belong, rather than to rebel [38].

These themes were evident to varying degrees in the health promotion materials. The health promotion materials in general had a strong focus on ‘engagement with family and community’ (Theme 1), ‘knowledge of risks of smoking,’ ‘giving up vs cutting down’ (Theme 2) and ‘culture in language and arts’ (Theme 3). ‘The impact of stress’ (Theme 1), ‘intrinsic beliefs and attitudes,’ and ‘limitations of health provider practice’ (Theme 2) were the themes that were least often incorporated into the health promotion materials. A critical reflection of the presence of this evidence in the health promotion materials is presented in Table 6 and forms the basis of the discussion and development of recommendations.

## 4. Discussion

This rapid review identified the extent to which peer-reviewed research evidence and health promotion materials for pregnant Aboriginal and Torres Strait Islander women who smoke reflect the domains of the Aboriginal Social and Emotional Wellbeing Model and the Behavior Change Wheel Model. Health promotion materials were then examined for the extent to which they reflected this evidence base. We present a critical reflection of these findings and make recommendations to inform the development of social media campaigns for smoking cessation amongst pregnant Aboriginal and Torres Strait Islander women in Australia and internationally.

### 4.1. Domains of Aboriginal Social and Emotional Wellbeing Model and Behavior Change Wheel Model

As expected, there was significant variance in the alignment of the domains of the Aboriginal Social and Emotional Wellbeing Model and the COM-B constructs in the research evidence and health promotion materials. While some topics raised by Aboriginal and Torres Strait Islander women within the conducted studies were well covered and highly aligned with the media messages, for example, the risks from smoking, issues related to quitting (giving up vs. cutting down) and engagement with family and community, other topics were infrequently portrayed as a topic within messages. Stress was reported as a factor related to smoking in pregnancy in 36% (4/11) peer-reviewed articles we assessed and was covered by only two health promotion campaigns (11%). The limitations of health provider practices, mentioned in 45% (5/11) of the papers, featured in 22% (4/18) campaigns. Despite ‘engagement with family and community’ being an important theme in the included papers, and featuring in most campaigns, few of the health promotion resources actually targeted family and community members alongside their focus on pregnant women. Culturally meaningful messages are vitally important to resonate with Aboriginal and Torres Strait Islander peoples. Culture was represented in the majority of resources vis language and the arts, but less prominent in terms of ‘knowing the past, dreaming of the future’ (used by about 50% of campaigns) and the focus on relationships (30%) and respect for community (30%).

It is acknowledged that when developing campaigns, pragmatic choices need to be made about the focus and content of messages. Media containing too many messages may be confusing. The campaign needs to be suitable for purpose, and co-design with Aboriginal and Torres Strait Islander communities is vital to promote ownership of messages and campaigns. It may not be necessary to cover all domains of the Aboriginal Social and Emotional Wellbeing Model or all components of the COM-B Model, although a holistic approach was taken by two campaigns which covered most domains of both models (*Stronger Boorais* and *Birthing in our Community)*. The Behavior Change Wheel and COM-B Model have been used as a toolkit to develop successful health promotion campaigns [25]. A behavioral diagnosis is helpful as a first stage when designing health promotional interventions. This approach has successfully been used for co-designing resources for pregnant Aboriginal and Torres Strait Islander women [56]. Appropriate intervention functions of the Behavior Change Wheel are usually chosen depending on what one aims to address in relation to the COM-B components [25]. In our review, most of the health promotion messages used education, persuasion and modelling however lesser used intervention functions that appeared in the evidence-based review, such as incentivization and enablement, could be candidates for future messages.

When using the Behavior Change Wheel and COM-B components, it is important to note that both physical and social opportunity are essential to maximize the effect of motivation and capability. Physical opportunities to quit smoking maybe lacking if health providers are not available or not skilled, and/or access to services is deficient. Social opportunities to quit smoking may be lacking if smoking is a social norm, there are few positive role models, or family and communities are not sufficiently supportive of pregnant women to quit smoking.

The themes and topics defined by this literature review and health promotion materials analyses may serve to broaden ideas for unique and less well-covered but relevant angles. Some issues raised by women within the context of research projects may be well-documented and broader due to rigorous designs to reduce bias. There are opportunities to use more of this evidence in media messages going forwards, whilst also concurrently offering these themes as considerations to communities when engaging in participatory processes of co-design.

Few research studies describe in detail the development of campaigns and how their health promotion messages were devised or the foundations for their messages. Aboriginal and Torres Strait Islander community-based workshops may be held with audience segmentation taken into account [57]. Resnicow et al. [58] recommend that culturally sensitive health promotion messages are developed on the foundation of surface and deep structure. Surface structure involves using messages that would be relevant to the outward characteristics of a target population. Deep structure takes account of the cultural, social, historical, environmental, and psychological factors that influence the target health behavior. Both surface and deep structure are important to ensure the fit and salience of messages, respectively. Deep structure aligns with many of the domains of the Aboriginal Social and Emotional Wellbeing Model related to family and kinship, community, country, spirit, spirituality, and ancestors. It also is relevant to our evidence-based themes and subthemes, especially the culturally meaningful theme.

In a survey of 47 organizations nationally about how anti-tobacco media messages were developed in Australia, Gould et al. [59] found that Aboriginal Medical Service were significantly more likely to report using deep structure in tailoring messages compared with non-government and government organizations. A dimension of “cultural understanding” evolved from a principal component analysis based on Aboriginal and Torres Strait Islander community engagement processes and the use of bottom-up approaches, community-based empowerment models and deep structure. Aboriginal organizations were more likely to excel in the use of this dimension [59].

Sinicrope et al. [60] described the protocol for the development of a social media intervention for Facebook to help Alaska Native people quit smoking based on cultural variance and surface/deep structure frameworks. The cultural variance framework considers the cultural influences on health behaviors in designing health messages including beliefs and norms, values, and Indigenous knowledge systems or ways of knowing [61]. The campaign used a co-design process based on digital storytelling with Alaska Native people who smoked and stakeholders. The phased, mixed methods approach evaluated existing media message content and co-develop test concepts for video and text-based messages and a Facebook group page. The videos, images and text are to be beta-tested through an online survey for cultural fit and impact along a range of indicators [60].

### 4.2. Strengths and Limitations of the Review

The cultural and methodological rigor embedded in our approach to the review and examination of evidence for social media campaign has not been done before for this particular population. Key components of this methodology included Aboriginal and Torres Strait Islander women as the focus of their studies who are the target audience of the included campaigns and honoring combined strengths of Aboriginal and non-Aboriginal researchers. We used strong theoretical frameworks from Aboriginal and Torres Strait Islander knowledges and behavioral and implementation science. The barriers and enablers to smoking cessation amongst pregnant Indigenous women have similarities internationally in high-income countries, although historical circumstances and social and cultural characteristics differ [62,63,64]. The methodology we developed here ensured that literature that only focused on Aboriginal and Torres Strait Islander women to ensure these women’s knowledge, attitudes and beliefs would inform the development of smoking cessation campaigns. This methodology the potential to inform the development and analysis of smoking cessation campaigns for pregnant First Nations women internationally.

There were a few limitations to the review. As a rapid review, not all the steps usually used in a systematic review were included. The search date was only until July 2019 and more recent peer-reviewed articles and campaigns may have been omitted. However, our findings will have immediate practical application in the development of an Australian Department of Health iSISTAQUIT social media campaign for pregnant Aboriginal and Torres Strait Islander women. These findings will guide the development of the community consultation and the media campaign to ensure that the topics explored are holistic according to the Aboriginal and Torres Strait Islander view of health and encompass the women’s requirements.

### 4.3. Recommendations for Research, Policy and Practice

There has been an increasing amount of evidence-based research over the last 10 years on the strengths and challenges of Aboriginal and Torres Strait Islander women who smoke in pregnancy. Future campaigns must draw on this burgeoning evidence base, and suitable frameworks may assist in this endeavor. LaVallee et al. [65] recommend that a suitable health research framework should reflect the interconnectedness and relationships between the individual and family, community, and larger environment, and recognize how these relationships influence the individual’s mental, physical, and spiritual health. Health is relational and must be addressed holistically. Both the Aboriginal Social and Emotional Wellbeing Model and the Behavior Change Wheel /COM-B are pragmatic frameworks to guide message focus. Combined, they take into account health and behavior from many angles. Using the current evidence plus a suitable framework, a contemporary co-design could lead to new, fresh yet salient campaign content.

Recommendations for research:Rigorous evaluation of a social media campaign; exploration of impact of incorporating domains of Aboriginal Social and Emotional Wellbeing Model in smoking cessation interventions.

Recommendations for policy:Incorporating equity and considerations of culture and cultural safety into funding and public health policy.

Recommendations for practice:Use an evidence-based approach that is equitable and culturally safe and owned by the local community—led by community members.

## 5. Conclusions

This rapid review of the literature was unique in synthesizing and analyzing peer-reviewed literature privileging Aboriginal and Torres Strait Islander women’s voices and then matching the results to existing media campaigns and messages for tobacco control and smoking cessation in pregnancy, using two well-respected frameworks of the Aboriginal Social and Emotional Wellbeing Model and the Behavior Change Wheel /COM-B Models. Our results found the following key themes from the peer-reviewed literature: holistic partnerships with women, their family, and their community; empowerment through knowledge and roles models; culturally meaningful.

The peer-reviewed literature frequently included the Aboriginal Social and Emotional Well-being domains of ‘Connection to Body’ and ‘Connection to Mind and Emotions’, ‘Connection to Family and Kinship’ and ‘Connection to Community’. Health promotion materials were more likely to incorporate ‘Connection to Culture’ and ‘Connection to Mind and Emotions’ than any other domain of the Aboriginal Social and Emotional Wellbeing Model. ‘Connection to Country’ was seldom represented in either the literature of health promotion materials and ‘Connection to Spirit, Spirituality and Ancestors’ was not in the literature and appeared on only one campaign.

Aspects of Behavior Change Wheel similarly well-covered in the literature and the health promotion materials were the COM-B constructs of psychological capability, and social and physical opportunity. Education was the intervention function most frequently featured in both the literature and health promotion materials. Whereas the health promotion materials also strongly featured intervention functions of persuasion and role modelling, the peer-reviewed studies were more likely to explore functions of enablement, followed by incentives and role modelling. The scope and content of the campaigns covered many of the emerging themes and framework components from the literature, but gaps were evident, especially related to a scarcity of appropriate messages targeted to families.

Additionally, interesting areas were revealed for future fresh approaches for health promotion and media messages. The methodology used in this review could be replicated to inform campaigns for international Indigenous women, minority populations, and those experiencing health disparities (adapted to local and country contexts).

## Figures and Tables

**Figure 1 ijerph-18-09341-f001:**
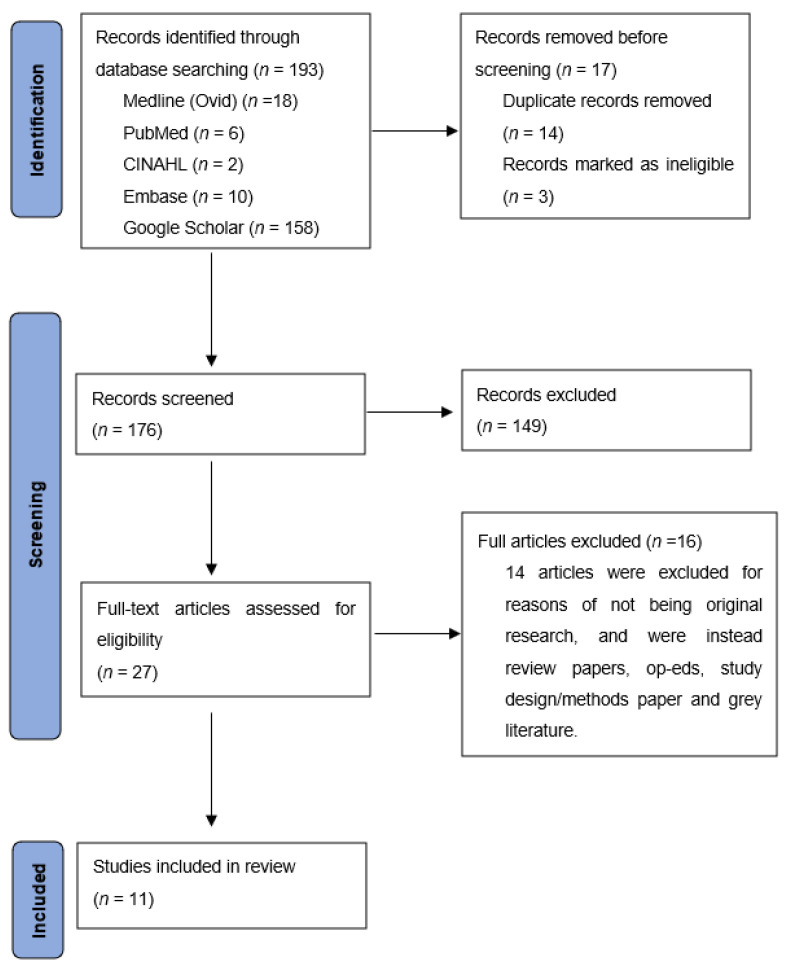
PRISMA flowchart. Adapted from [26].

**Table 1 ijerph-18-09341-t001:** Search strategy.

Component	Search Method	Search Strategy
Peer Reviewed Literature	Database	Ovid/MEDLINE, Embase, CINAHL, PubMed, Google Scholar
	Keyword	Pregnancy AND (health promotion OR (health messages OR health knowledge OR attitudes OR practices)) AND (Aboriginal OR Aboriginal and Torres Strait Islander OR Indigenous Australians OR Oceanic Ancestry Group AND Australia) AND (smoking OR smoking cessation)
	Inclusion criteria	Publications from 2008 to 2019 Women pregnant with an Aboriginal and/or Torres Strait Islander baby who smoke, Australia, original research
	Exclusion criteria	Published pre-2008, not based in Australia, not exclusive to women pregnant with an Aboriginal and/or Torres Strait Islander baby, grey literature, not original research (e.g., reviews, opinion pieces)
Media Campaigns *	Social Media and Online Platforms	Facebook, YouTube, Vimeo, Google
	Search terms	Combinations of Tackling Indigenous Smoking, Aboriginal, Indigenous, smoking, quit smoking, pregnancy
	Inclusion Criteria	Health promotion materials sourced from 2008 to 2019 Smoking cessation information for pregnant Aboriginal and/or Torres Strait Islander women and their families (print brochures, posters, video, social media)
	Exclusion criteria	Not targeted at pregnant Aboriginal nor Torres Strait Islander women, published pre-2008 Media resources used only in a research context (e.g., trial, pilot study)

* Additional media campaigns were obtained through snowball technique.

**Table 2 ijerph-18-09341-t002:** Summary of health promotion materials and summary characteristics of included articles.

Author (Date)	Aim	Methods (Setting, Design, Participants)	Outcomes
Askew et al. (2019) [33]	Determine the impact of the *Empowering Strong Families 4077* smoking cessation intervention on smoking rates of pregnant women and their significant others and determine its feasibility and acceptability to study participants and their primary health care service.	Urban. Semi-structured interviews (*N* = 17) with pregnant Aboriginal women who smoke (*n* = 7), significant others (*n* = 6), and health providers (*n* = 4)	High levels of satisfaction from participants and health professionals. Valued that the approach was flexible, strength-based, family-centered, holistic, relationship-based, and participant-led. Case management facilitated disclosure of complexities in personal life which had not otherwise been disclosed, facilitating access to relevant supports.
Bovill et al. (2017) [34]	To privilege the voices of Aboriginal women, collecting their experiences of smoking during pregnancy and smoking cessation care, to collaborate and find effective solutions.	Unclear if urban or rural. In-depth interviews with pregnant or recently pregnant Aboriginal women (*N* = 20)	Themes of barriers to accepting smoking cessation support: (i) Ambivalence to a need for support; (ii) Health professional advice; (iii) Reduction in smoking; (iv) Attitudes towards nicotine replacement therapy; (v) Suggested strategies
Bovill et al. (2018) [35]	Understand what educational resources are needed to support quit smoking attempts to inform an evidence-based intervention	Unclear if urban or rural. Yarning circles (*N* = 24) with a pregnant Aboriginal woman (*n* = 1), mothers (*n* = 15), and elders (*n* = 8)	Predetermined Themes: (i) Graphics and layout influenced attraction; (ii) Comprehension; (iii) Persuasion and self-efficacy; (iv) Cultural acceptability Emergent Themes: (i) Make resources more interactive; (ii) Tell me more; (iii) Non-pharmacological approaches
Gilligan et al. (2009) [27]	Identify predisposing, enabling, and reinforcing factors associated with smoking among pregnant Aboriginal and Torres Strait Islander women.	Remote and regional. Cross-sectional quantitative surveys with pregnant Aboriginal women (*N* = 145)	Enabling/reinforcing factors for smoking were having a partner who smoked and high or very high levels of daily stress
Gould et al. (2013) [36]	Explore attitudes and experiences related to prenatal smoking by Aboriginal women and household smoking, provide recommendations for culturally appropriate interventions.	Regional. Focus groups with pregnant Aboriginal women and family members (*N* = 18)	Themes: (i) Social and family influences on smoking; (ii) Knowing and experiencing the health effects from smoking; (iii) Responses to health messages; (iv) Managing smoke-free homes and cars; (v) Stress and craving; (vi) Giving up and cutting down; (vii) Community recommendations
Gould et al. (2017) [37]	Explore Aboriginal women’s narratives from starting smoking through to pregnancy.	Unclear if urban or rural. In-depth interviews with pregnant or recently pregnant Aboriginal women (*N* = 20)	Themes: (i) The role mothers play in women’s smoking and quitting; (ii) The contribution of nausea to spontaneous quitting; (iii) Depression as a barrier to quitting; and (iv) The hopes of women for their own and their children’s future. The epiphany of pregnancy was a key turning point for many—including the interplay of successive pregnancies; and the intensity of expressed regret.
Passey et al. (2011) [38]	Explore factors contributing to smoking initiation among rural Aboriginal women and girls and the social context.	Rural. In-depth interviews and focus groups (*N* = 36) with pregnant or recently pregnant Aboriginal women (*n* = 22); and health providers (*n* = 14)	Themes: (i) Colonization and the introduction of tobacco; (ii) Normalization of smoking within separate Aboriginal social networks; (iii) Disadvantage and stressful lives; and (iv) The importance of maintaining relationships within the extended family and community networks.
Passey et al. (2012) [29]	To compare the knowledge and attitudes of those who smoke and don’t smoke during pregnancy; and those who quit smoking, and those who continue during pregnancy	Remote and regional. Cross-sectional quantitative surveys with pregnant Aboriginal women who smoke, and who had never smoked (*N* = 149)	Women who stopped smoking had greater knowledge of smoking risks. Women who did not stop smoking had stronger attitudes that smoking is acceptable and quitting is difficult
Passey, Sanson-Fisher & Stirling (2014) [28]	Assess support for 12 potential smoking cessation strategies among pregnant Australian Indigenous women	Remote and regional. Cross-sectional quantitative surveys with pregnant Aboriginal women who smoke (*N* = 121)	The proportion of women who rated the strategies very or somewhat helpful: Support for the whole family (64%); Rewards (63%); Advice and support (midwife) (62%); Advice and support (doctor) (61%); Community activities (59%); Advice and support (Aboriginal Health Worker) (56%); Free nicotine replacement therapy (56%); Peer support groups (53%); Brochures (52%); Stress management programs (49%); Support person (47%); Quitline (46%)
Passey & Stirling (2018) [39]	Assess the acceptability of *Stop smoking in its tracks* to women and providers and the feasibility of implementation	Rural. Semi-structured interviews with pregnant Aboriginal women who smoke (*N* = 13)	The women appreciated the frequency of support, the information provided, household support and free nicotine replacement therapy (for the family as well). Rewards were motivating and helpful. The women enjoyed support groups.
Wood et al. (2008) [40]	Investigate knowledge, cultural contexts, and barriers to smoking cessation for pregnant Aboriginal women	Urban. Focus groups and in-depth interviews (*N* = 40) with Aboriginal women (mostly mothers) who were pregnant (*n* = 7), smoke (*n* = 30), or had quit smoking (*n* = 7); and Aboriginal Health Workers (*n* = 10)	Themes: (i) Context of smoking, and continuing to smoke during pregnancy; (ii) Positives and negatives of smoking; (iii) Pregnancy as a catalyst for a change in smoking behavior; (iv) Awareness of the risks related to smoking during pregnancy; (v) Smoking reduction during pregnancy; (vi) Quitting smoking; (vii) Strategies for quitting smoking; (viii) First time pregnancies: a time to intervene; (ix) The salience of passive smoking; (x) The role of Aboriginal Health Workers

**Table 3 ijerph-18-09341-t003:** Summary of health promotion materials.

Campaign (Source) Platform	Content Summary	Audience	Calls to Action	Frame (Loss/Gain)	Appeal (Threat/Efficacy)	Centricity (Infant/Maternal)
**Video resources**						
Strong Boorais, Bright Futures (Alcohol and Drug Foundation) [41] YouTube; CD	A short film encouraging pregnant Aboriginal women to quit smoking and engage with a variety of health providers. Aboriginal women encouraging women ‘don’t be shame,’ yarn with someone they trust, and return to country and family support	Pregnant women	Stop smoking; engage with health providers; seek family and social support; adopt a healthy lifestyle; return to country	Both	Efficacy	Both
No Smokes [42] Website; Facebook; Instagram	Very short animation with male narrator outlining the consequence of smoking when pregnant, including animation of a coffin in utero. An Aboriginal woman talks to the camera about her quit journey and the impact on the baby	Pregnant women	Talk to a health worker about quitting; Quitline	Loss	Threat	Infant
Smoking: They Can’t Choose, It’s Up To You (Central Australian Aboriginal Congress) [43] YouTube; Facebook	Very short film including imagery of a fetus coughing out smoke, a pregnant woman putting down smokes, smiling, and walking on, and a black and white clip of family and friends smoking at a barbecue. The clip then replays in color with nobody smoking	Pregnant women	Talk to Congress about quitting	Gain	Both	Both
Deadly Choices (Institute for Urban Indigenous Health) [32] TV ads, YouTube; Facebook, Instagram	Two TV Ads: (i) Pregnant mum is preparing dinner while her children do their homework. She goes outside to ‘give Nanny a call’ and lights a cigarette. Her daughter comes out and tells her to come inside. The mum puts the smokes in the bin. Voiceover: ‘I can’t remember why I started smoking, but I’ll remember why I stopped.’ (ii) Pregnant woman sits down to have a smoke and reads a letter from her deadly daughter talking about what a great mum she is, and that she saw a photo of what her baby sister might look like. The photo is a sick baby from a cigarette packet. The woman puts her smokes in the bin and walks off. Voiceover: ‘I can’t remember why I started smoking, but I’ll remember why I stopped.’	Pregnant women	Stop smoking; contact local Aboriginal Medical Service; Quitline	Gain	Both	Infant
Don’t Make Smokes Your Story (Apunipima Cape York Health Council) [44] YouTube; Facebook	Short film about the harms of smoking in pregnancy. Pregnant woman starts to pick up smokes but picks up health brochure instead. Educational voiceover. The woman walks out and into her local health service for a consultation. Returns home smiling and puts smokes in bin.	Pregnant women	Stop smoking; see a health provider; nicotine replacement therapy	Loss	Efficacy	Both
Butt Out Boondah (Grand Pacific Health) [45] YouTube; Facebook	Short film with pregnant woman talking to the camera about how she always knew the risks to her but is now concerned for her baby and wants what is best for baby. Talks about the high smoking prevalence in her region. Says ‘If I can do it, you can too, because you’re not the only one it’s affecting.’	Pregnant women	Stop smoking	Both	Efficacy	Both
Ready Mob Smoke Free Community (Galambila Aboriginal Medical Service) [31] YouTube; Facebook; Instagram; Website	Short film with a health worker talking about the risks of smoking for baby.	Pregnant women	See Quit for New Life	Loss	Efficacy	Infant
Indigenous Mothers Talk (Townsville Aboriginal and Islander Health Service) [46] YouTube	Short film with a health worker talking to two women—one has successfully quit smoking, the other is still smoking. She does a demonstration with Smokey Suzie and talks about the impact of smoking on the fetus. The women talk about barriers to cessation (for example: stress, partners and household members who smoke); women appreciated the education and support from the health service.	Pregnant women	Go to someone who can help you quit	Loss	Efficacy	Both
Puyu paki (Don’t smoke—give it up) (Puntukurnu Aboriginal Medical Service) [47] YouTube; Facebook	Two films: (i) Pregnant woman smoking in a playground. The voiceover is in language, with text in English (Mum smoked), Grandad smoking on the couch next to a child (Grandad smoked), Dad smoking in car with teenager pressed against the glass (Dad smoked), teenager in hospital on oxygen (now, I’m sick). (ii) Pregnant Aboriginal woman looking in the mirror, then image of smoke going to fetus. Voiceover and text say smoking costs more than just money and details the effects on bub. Then sunny day, older Aboriginal woman laughing and watching older children playing in park.	Women Men (fathers and grandfathers)	Quit smoking; contact Puntukurnu Aboriginal Medical Service	Both	Both	Infant
Blow Away the Smokes (Mid North Coast Division of General Practice) [48] Vimeo; CD	A 30-min film targeting all Aboriginal people who smoke, with a brief section for pregnant women. This section has smoking advice provided by an Aboriginal Obstetrician, as well as a role model (woman who quit smoking with a child in a pram).	Aboriginal people who smoke—one section is dedicated to pregnancy	Quit smoking; nicotine replacement therapy	Both	Efficacy	Infant
Ngamari Free (WA Country Health Service) [49] Brochure; animated clip; billboard	Animated film of a pregnant Aboriginal woman in consultation with a male, non-Aboriginal doctor. He says he has seen her smoking in the carpark and provides information on the harms to baby. Clip ends with mum happily giving birth to a full-term baby with her supportive partner there. Brochures provide information on the variety of support services available and the risks to bub.	Pregnant women	Quit smoking; contact Wheatbelt Aboriginal Health Service and Public Health Unit; Quitline; contact health providers; Quit for you—Quit for Two app	Both	Efficacy	Both
Bega Garnbirringu Health Service [50] Poster, film clip	Short film clip of Aboriginal comedian talking to camera about the importance of quitting for pregnancy and the risks to baby. Poster with illustration of smoke reaching fetus and the harms it can cause.	Pregnant women	Quit smoking; smoke safely; talk to Bega Garnbirringu Health Service; Quitline	Both	Both	Infant
**Non-video resources**						
‘Mary G’ Campaign (Australian Council of Smoking and Health) [51] Radio slots; flyer; podcast	A series of short radio commercials by Aboriginal comedy personality ‘Mary G’, with straight-up messages about the harm smoking does to a baby and the need to quit smoking.	Women	Quit smoking; don’t smoke around children or when pregnant	Loss	Fear	Infant
Quit For New Life (NSW Health) [52] Brochures	Brochures for the NSW Health Quit for New Life Campaign—messages include ‘Your baby needs you to quit’ and ‘Your baby needs you to be a strong and healthy mum.’ Strong focus on Aboriginal artwork, information on the risks of smoking, and the benefits of quitting	Pregnant women	Quit smoking; talk to a health provider; Quitline	Gain	Efficacy	Both
Stickin’ it up the Smokes (Aboriginal Health Council of South Australia) [30] Facebook, posters, song + film clip, door hanger, bumper sticker, ‘Bump to Bub’ booklet, Flip Chart	A multifaceted campaign with punk-cool pregnant women in the posters and simple messages to quit smoking. The Bump to Bub booklet features healthy ‘glowing’ pregnant women and provides information on pregnancy, fetal growth, and smoking cessation over a 9-month period. The children born to the Bump to Bub mothers went on to star in a revised campaign.	Pregnant women	Speak to a health provider; Quitline; quit smoking; contact local Aboriginal Health team; visit campaign website	Gain	Efficacy	Infant
Want the best for your baby? (Quitline Tasmania) [53] Flyer	Flyer with photograph of hands cradling a pregnant belly with Aboriginal artwork in the background. Text focuses on the benefits for mum + bub that quitting brings and the 4Ds (Delay, Deep breathe, Do something else, Drink water)	Pregnant women	Contact Quitline	Gain	Efficacy	Both
Birthing in our Community (Institute for Urban Indigenous Health) [54] Brochure	Two brochures—one for pregnant women, one for others in her life on risks for bub and how they can support her to quit. Photographs of pregnant bellies, happy babies, and sick babies. Education on harms of smoking for bub, benefits of quitting for mum and bub, the 4Ds (Delay, Deep breathe, Do something else, Drink water)	Pregnant women, family and friends	Contact health provider	Both	Both	Both

**Table 4 ijerph-18-09341-t004:** Alignment of peer reviewed studies with the domains of Aboriginal Social and Emotional Wellbeing Model and Behavior Change Wheel.

Paper 1st Author	Aboriginal Social & Emotional Well-Being							Capability		Opportunity		Motivation			Intervention Functions							
	B	M&E	F&K	Cou	Com	Cul	S	Ps	Ph	So	Ph	Re	Au	Pe	Ed	Tr	Co	Er	En	In	Re	Mo
Askew (2019) [33]	✓	✓			✓	✓		✓	✓	✓	✓	✓							✓	✓		
Bovill (2017) [34]	✓	✓	✓		✓			✓		✓	✓	✓	✓		✓				✓			
Bovill (2018) [35]	✓	✓			✓	✓		✓		✓	✓	✓	✓		✓							✓
Gilligan (2009) [27]	✓	✓						✓		✓												
Gould (2013) [36]	✓	✓	✓		✓			✓	✓	✓	✓	✓	✓	✓	✓							
Gould (2017) [56]	✓	✓	✓					✓		✓	✓	✓	✓	✓					✓			✓
Passey (2011) [38]	✓	✓	✓	✓	✓			✓		✓		✓	✓									✓
Passey (2012) [29]	✓	✓			✓			✓		✓		✓			✓							
Passey (2014) [28]	✓	✓	✓		✓			✓	✓	✓	✓				✓				✓	✓		
Passey (2018) [39]	✓	✓	✓		✓			✓	✓	✓	✓	✓			✓				✓	✓		
Wood (2008) [40]	✓	✓	✓		✓			✓	✓	✓	✓	✓	✓		✓				✓		✓	

Note: Body (B), Mind and Emotions (M&E), Family and Kinship (F&K), Country (Cou), Community (Com), Culture (Cul), Spirit, spirituality and ancestors (S); Psycho-logical (Ps), Physical (Ph), Social (So), Physical (Ph), Reflective (Re), Automatic (Au), Persuasion (Pe), Education (Ed), Training (Tr), Coercion (Co), Environment restructuring (Er), Enablement (En), Incentivization (In), Restrictions (Re), Modelling (Mo).

**Table 5 ijerph-18-09341-t005:** Alignment of health promotion materials with the domains of Aboriginal Social and Emotional Wellbeing Model and Behavior Change Wheel.

Campaign	Aboriginal Social and Emotional Well-Being							Capability		Opportunity		Motivation			Intervention Functions							
	B	M&E	F&K	Cou	Com	Cul	S	Ps	Ph	So	Ph	Re	Au	Pe	Ed	Tr	Co	Er	En	In	Re	Mo
Strong Boorais [41]	✓	✓	✓	✓	✓	✓		✓	✓	✓	✓	✓	✓	✓	✓							✓
No Smokes [42]	✓							✓				✓		✓	✓							✓
It’s Up To You [43]	✓	✓	✓		✓	✓	✓	✓		✓	✓	✓	✓	✓	✓							✓
Deadly Choices [32]	✓	✓	✓					✓		✓	✓	✓	✓	✓								✓
Don’t Make Smokes [44]	✓					✓		✓	✓		✓	✓	✓	✓	✓							✓
Butt Out Boondah [45]	✓							✓		✓	✓	✓	✓	✓	✓							✓
Ready Mob [31]	✓							✓		✓	✓	✓			✓							
Mothers Talk [46]	✓	✓						✓	✓	✓	✓			✓	✓							✓
Puyu paki [47]	✓	✓	✓		✓	✓		✓		✓	✓	✓	✓	✓	✓							
‘Mary G’ [51]	✓					✓		✓				✓		✓	✓							
Blow Away Smokes [48]	✓	✓				✓		✓			✓	✓		✓	✓							✓
Quit For New Life [52]	✓	✓	✓			✓		✓	✓	✓	✓	✓	✓	✓	✓							
Stickin’ it up Smokes [30]	✓		✓					✓		✓		✓	✓	✓	✓							✓
Ngamari Free [49]	✓	✓	✓		✓	✓		✓	✓	✓	✓	✓	✓	✓	✓							✓
Bega Garnbirringu [50]	✓							✓				✓	✓	✓	✓							
Want the best for your baby? [53]	✓	✓				✓		✓	✓		✓	✓		✓	✓							
Birth in Community [54]	✓	✓	✓	✓	✓	✓		✓	✓	✓	✓	✓	✓	✓	✓							

Note: Body (B), Mind and Emotions (M&E), Family and Kinship (F&K), Country (Cou), Community (Com), Culture (Cul), Spirit, spirituality and ancestors (S); Psychological (Ps), Physical (Ph), Social (So), Physical (Ph), Reflective (Re), Automatic (Au), Persuasion (Pe), Education (Ed), Training (Tr), Coercion (Co), Environment restructuring (Er), Enablement (En), Incentivization (In), Restrictions (Re), Modelling (Mo).

**Table 6 ijerph-18-09341-t006:** Alignment of messages within the health promotion materials with the peer reviewed literature themes.

Campaign	Holistic Partnerships with Women, Their Family, and Their Community				Empowerment through Knowledge and Role Models						Culturally Meaningful		
	NRT	Mechanism of Support	Impact of Stress	Engagement with Family and Community	Intrinsic Beliefs and Attitudes	Knowledge of Risks of Smoking	Giving Up vs. Cutting Down	Limitations of Health Provider Practice	A woman’s Autonomy and the Role of Family and Community	Knowing the Past, Dreaming of the Future	Culture in Language and Arts	Relationships with Community	Respect for Relatives & Aboriginal and Torres Strait Islander Community
Strong Boorais [41]		✓	✓	✓	✓	✓	✓	✓	✓	✓	✓	✓	✓
No Smokes [42]						✓	✓						
It’s Up To You [43]				✓		✓	✓				✓	✓	✓
Deadly Choices [32]				✓		✓	✓		✓	✓	✓		
Don’t Make Smokes [44]	✓					✓	✓				✓		
Butt Out Boondah [45]				✓		✓	✓		✓				
Ready Mob [31]		✓		✓		✓	✓		✓		✓		✓
Mothers Talk [46]	✓	✓	✓	✓	✓	✓	✓		✓			✓	
Puyu paki [47]				✓		✓	✓		✓	✓	✓	✓	
‘Mary G’ [51]						✓	✓				✓		
Blow Away Smokes [48]	✓	✓		✓	✓	✓	✓			✓			✓
Quit For New Life [52]		✓		✓		✓	✓		✓	✓	✓		
Stickin’ it up Smokes [30]				✓					✓	✓	✓		
Ngamari Free [49]	✓	✓		✓		✓	✓	✓			✓		
Bega Garnbirringu [50]						✓	✓				✓	✓	✓
Want the best for your baby? [53]					✓	✓	✓	✓			✓		
Birth in Community [54]	✓	✓		✓	✓	✓	✓	✓	✓	✓	✓	✓	✓

## Data Availability

Not applicable.

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
