# Peer review of "Smoking Cessation Messages for Pregnant Aboriginal and Torres Strait Islander Women: A Rapid Review of Peer-Reviewed Literature and Assessment of Research Translation of Media Content"

_ijerph, 2021, doi:10.3390/ijerph18179341_

Round 1

Reviewer 1 Report

The authors have adequately addressed my previous comments and suggestions

Author Response

Thank you to the reviewer for confirming the authors have adequately addressed previous comments and suggestions. The authors completed a spell check to address minor spelling issues. 

Reviewer 2 Report

The Authors prepared extensive revisions and all the comments were addressed correctly. The scientific soundness of this paper has increased. 

Author Response

Thank you to the reviewer for confirming the authors have adequately addressed previous comments and suggestions. The authors completed a spell check to address minor spelling issues. 

This manuscript is a resubmission of an earlier submission. The following is a list of the peer review reports and author responses from that submission.

Round 1

Reviewer 1 Report

Thank you for inviting me to review this publication.

The introduction was particularly well written, and provided a robust rationale for the study. I congratulate the authors and thank them for the time they took on the introduction. It was a pleasure to read.

Papers were included if they were published from 2008 to 2019, with the rationale for the time frame being the introduction of the Australian government’s program Tackling Indigenous Smoking. However, the study by Wood et al & Gilligan et al were published in 2008 & 2009 respectively, but the research occurred before this date and is therefore unrelated to the government program. The authors should either reconsider the timeframe for the included studies, or revise their rationale. Was the search rerun to include any relevant papers published in 2020 and the first half of 2021?

The authors appear to be using ‘Aboriginal’ as being inclusive of Torres Strait Islander people in some locations in the paper. This needs to be made explicit, or more preferably, Aboriginal and Torres Strait Islander written in each instance it is implied.

The data were extracted by one author. What strategies were employed to ensure accuracy in this critical step? Similarly, decisions about inclusion and exclusion were made by one author -

Quantitative studies were categorized in accordance with relevant domains of the BCW, COM-B and Aboriginal Social and Emotional Wellbeing Model frameworks. How were qualitative or mixed methods studies categorised? How were judgements made during this categorisation? What strategies were employed to ensure consistency and rigour in these judgements.

The Aboriginal and Torres Strait Islander Social and Emotional Wellbeing Model was used to classify the literature and health promotion materials, but no justification is given for the use of this model and I am unconvinced of the appropriateness or the utility of this decision. SEWB, as presented by Gee et al (the cited reference), is “a multidimensional concept of health that includes mental health, but which also encompasses domains of health and wellbeing such as connection to land or ‘country’, culture, spirituality, ancestry, family, and community”. Gee et al proposed the domains, not as a definitive list or model for evaluation of health promotion programs, but rather as some of the domains of wellbeing that typically characterise Aboriginal and Torres Strait Islander definitions of SEWB. These issues need to be addressed.

Did the authors consider any other framework for classifying the included studies and health promotion materials, for example, Megan Williams’ Ngaa-bi-nya Aboriginal and Torres Strait Islander program evaluation framework may be more suitable for the health promotion materials (Williams M. Evaluation Journal of Australasia 2018;18:6-20).

The thematic review of the published literature adds little to existing body of literature reviews on related topics and I am not convinced that it is necessary, particularly as there is a lack of methodological rigour in this element of the review. For example, more detail is required about how the thematic analysis was done and how the second researcher verified the analysis and results?

194 articles were identified. 182 were excluded for various reasons. This leaves 12 studies, but the paper reports 11 studies were included – this needs to be corrected or clarified. The total number of potential media campaigns initially identified needs to be reported, and how the final list of 17 campaigns were determined. Characteristics of the included studies and campaigns are presented as one table in the supplementary materials – I believe that this table should be included in the text as it provides the reader with essential information.

The authors state that all but 3 included studies were qualitative studies and the “quantitative studies were included because they had Aboriginal and/or Torres Strait Islander women as participants and provided quantitative data to complement qualitative findings”. Does this mean that these 3 studies employed both qualitative and quantitative? If this is so, then the authors should state that they were mixed methods studies. The type of research was not list previously as an inclusion or exclusion criteria but it would appear that this was the case. Can this please be clarified. The mixed methods studies are inaccurately reported in the table of characteristics, with only the qualitative component being reported. This is not acceptable, and needs to be corrected, or it made clear to the reader that only the qualitative data is being reported.

The included studies and campaigns need to be referenced in the tables to enable the reader to identify the paper quickly and easily in the reference list.

On a minor issue, there are a number of minor typos need to be corrected prior to publication, eg. 1st sentence on pg 4 (line 147) – either a connecting word is required between ‘account’ and ‘this’, or the comma should be changed to a colon or fullstop. Line 148 - I would suggest that ‘focus’ be changed to focused.  Line 52, ‘is targeting’ should be changed to ‘has been targeting’. Please check the bracketing in the search strategy presented in Table 1 – there appears to be a ‘close bracket’ missing.

In summary, this paper could make a valuable contribution but the above issues need to be addressed before the journal should consider publishing it.

Author Response

5rd August, 2021 

RE: [IJERPH] Manuscript ID: ijerph-1261445 - Major Revisions

Dear Editor and reviewers,

Thank you for reviewing the manuscript No. IJERPH-1261445. Please find below our response addressing each of the raised items identified in the major revisions letter.

REVIEWER 1

Open Review

The introduction was particularly well written, and provided a robust rationale for the study. I congratulate the authors and thank them for the time they took on the introduction. It was a pleasure to read.

Papers were included if they were published from 2008 to 2019, with the rationale for the time frame being the introduction of the Australian government’s program Tackling Indigenous Smoking. However, the study by Wood et al & Gilligan et al were published in 2008 & 2009 respectively, but the research occurred before this date and is therefore unrelated to the government program. The authors should either reconsider the timeframe for the included studies, or revise their rationale. Was the search rerun to include any relevant papers published in 2020 and the first half of 2021?

We were looking for evidence in the literature from 2008 onwards. In terms of rerunning the papers published in 2020, we didn’t have the resources to update the review.

The authors appear to be using ‘Aboriginal’ as being inclusive of Torres Strait Islander people in some locations in the paper. This needs to be made explicit, or more preferably, Aboriginal and Torres Strait Islander written in each instance it is implied.

  • The term “Aboriginal and Torres Strait Islander” is used throughout the text with exception of the quotes of other texts, specific terminology such as Aboriginal Medical Service or methods that are defined as Aboriginal.

The data were extracted by one author. What strategies were employed to ensure accuracy in this critical step? Similarly, decisions about inclusion and exclusion were made by one author –

  • Compared to systematic reviews, rapid reviews aim to be finished in less than 6 months and have excluded some steps, for example not using two reviewers for the study selection, not conducting a quality assessment of the studies and including limited sources for the review.

Quantitative studies were categorized in accordance with relevant domains of the BCW, COM-B and Aboriginal Social and Emotional Wellbeing Model frameworks. How were qualitative or mixed methods studies categorised? How were judgements made during this categorisation? What strategies were employed to ensure consistency and rigour in these judgements.

  • This was an error in the text. Qualitative and quantitative studies were categories in the same way. The information has been corrected.

The Aboriginal and Torres Strait Islander Social and Emotional Wellbeing Model was used to classify the literature and health promotion materials, but no justification is given for the use of this model and I am unconvinced of the appropriateness or the utility of this decision. SEWB, as presented by Gee et al (the cited reference), is “a multidimensional concept of health that includes mental health, but which also encompasses domains of health and wellbeing such as connection to land or ‘country’, culture, spirituality, ancestry, family, and community”. Gee et al proposed the domains, not as a definitive list or model for evaluation of health promotion programs, but rather as some of the domains of wellbeing that typically characterise Aboriginal and Torres Strait Islander definitions of SEWB. These issues need to be addressed.

  • The intention of SEWB was not to evaluate the program but to analyse the focus and scope of the literature and health messages that have been previously used.

We have added in the following paragraph:

The Aboriginal and Torres Strait Islander Social and Emotional Wellbeing Model was used as it proposes a multidimensional view of health from an Aboriginal and Torres Strait Islander standpoint. It was a framework that was developed to help clinicians address the uncertainty in their work about how to approach Aboriginal and Torres Strait Islander Health [23]. Consequently, the model clarifies guiding principles and broad areas of wellbeing that need to be considered when working in this field with Aboriginal and Torres Strait Islander peoples. While not intended by the originators as a definitive model or way to classify literature or health promotion programs, our exploratory use for this purpose, we believe, is warranted. The model was used pragmatically to help explore the focus and scope of messages related to an Aboriginal and Torres Strait Islander holistic concept of health, rather than a model to assess a program or intervention.

Did the authors consider any other framework for classifying the included studies and health promotion materials, for example, Megan Williams’ Ngaa-bi-nya Aboriginal and Torres Strait Islander program evaluation framework may be more suitable for the health promotion materials (Williams M. Evaluation Journal of Australasia 2018;18:6-20).

  • We are aware of the Ngaa-bi-nya Aboriginal and Torres Strait Islander program evaluation framework but it did not fit our purpose for this review.

The thematic review of the published literature adds little to existing body of literature reviews on related topics and I am not convinced that it is necessary, particularly as there is a lack of methodological rigour in this element of the review. For example, more detail is required about how the thematic analysis was done and how the second researcher verified the analysis and results?

  • The information has been corrected, now it says An inductive approach to analysis was adopted, with TF conducting coding and initial theme generation. This was then reviewed with an Aboriginal researcher (AH), by assessing the themes for alignment with the literature and Aboriginal cultural meanings. In this step, AH utilized cultural knowledges and interpretations of the literature to the initial themes, applying an additional layer of understanding and meaning to the analysis. The themes were then refined and named in accordance with their eventual definitions.”

194 articles were identified. 182 were excluded for various reasons. This leaves 12 studies, but the paper reports 11 studies were included – this needs to be corrected or clarified. The total number of potential media campaigns initially identified needs to be reported, and how the final list of 17 campaigns were determined. Characteristics of the included studies and campaigns are presented as one table in the supplementary materials – I believe that this table should be included in the text as it provides the reader with essential information.

  • Thanks for the correction, the identified articles were 193. We have adjusted the text to express this.
  • The annex table has been included in the text. In terms of the campaigns, the search hasn’t presented the same steps of the rapid review, as there is not a standardized method for reviewing media campaigns.

The authors state that all but 3 included studies were qualitative studies and the “quantitative studies were included because they had Aboriginal and/or Torres Strait Islander women as participants and provided quantitative data to complement qualitative findings”. Does this mean that these 3 studies employed both qualitative and quantitative? If this is so, then the authors should state that they were mixed methods studies. The type of research was not list previously as an inclusion or exclusion criteria but it would appear that this was the case. Can this please be clarified. The mixed methods studies are inaccurately reported in the table of characteristics, with only the qualitative component being reported. This is not acceptable, and needs to be corrected, or it made clear to the reader that only the qualitative data is being reported.

  • Papers did not need to have a qualitative component to be included. Purely quantitative papers could be included as long as they had Aboriginal and/or Torres Strait Islander women as participants and addressed the research aim.

The included studies and campaigns need to be referenced in the tables to enable the reader to identify the paper quickly and easily in the reference list.

  • We have included the references

On a minor issue, there are a number of minor typos need to be corrected prior to publication, eg. 1st sentence on pg 4 (line 147) – either a connecting word is required between ‘account’ and ‘this’, or the comma should be changed to a colon or fullstop. Line 148 - I would suggest that ‘focus’ be changed to focused.  Line 52, ‘is targeting’ should be changed to ‘has been targeting’. Please check the bracketing in the search strategy presented in Table 1 – there appears to be a ‘close bracket’ missing.

  • We have included the changes.

Reviewer 2 Report

The presented review is novel and interesting. However, some minor revisions are needed:

  1. The authors should clearly define why do the study carried out among Aboriginal and Torres Strait Islander Australians is important for tobacco control research - please provide 2-3 sentences in the Introduction that may be informative for the international (non-Australian) readers.
  2. According to the reviewer's opinion citation presented in lines 34-36 seems to be inaccurate in the case of scientific papers.
  3. The authors refer to part 1 and part 2 (line 128) please clearly define what part 1 and part 2 mean (which objectives - part 1  -  study on..)
  4. Table 1 is very informative. Thank you for providing a well-prepared and clearly defined  methodology 
  5. Please consider adding Prisma Flow Chart
  6. Please clearly define practical implications of this study

Author Response

5rd August, 2021

RE: [IJERPH] Manuscript ID: ijerph-1261445 - Major Revisions

Dear Editor and reviewers,

Thank you for reviewing the manuscript No. IJERPH-1261445. Please find below our response addressing each of the raised items identified in the major revisions letter.

Reviewer 2

  1. The authors should clearly define why do the study carried out among Aboriginal and Torres Strait Islander Australians is important for tobacco control research - please provide 2-3 sentences in the Introduction that may be informative for the international (non-Australian) readers.

  • In Australia, Aboriginal and Torres Strait Islander women have higher smoking rates compared with non-Aboriginal women, observing a percentage of 44% compared with 11%[2]. Therefore, promoting a smoke-free pregnancy benefits Aboriginal and Torres Strait Islander women and their babies, including reducing the risk of perinatal death, preterm birth, small for gestational age, and severe neonatal morbidity [1]. While Aboriginal and Torres Strait Islander people are more likely to attempt to quit smoking than other Australians, they have less success [3]; specifically, during pregnancy, Aboriginal and Torres Strait Islander women are highly motivated to quit smoking, but they are widely unsupported to do so by health providers [4].

  1. According to the reviewer's opinion citation presented in lines 34-36 seems to be inaccurate in the case of scientific papers.

  • The citation has been corrected and read as [1].

  1. The authors refer to part 1 and part 2 (line 128) please clearly define what part 1 and part 2 mean (which objectives - part 1  -  study on..)

  • Part 1 and 2 have been included in the aims.

  1. Table 1 is very informative. Thank you for providing a well-prepared and clearly defined methodology 

  • The table has been included in the main text.

  1. Please consider adding Prisma Flow Chart.

  • The Prisma Flow Chart has been included in the text.

  1. Please clearly define practical implications of this study

  • The following information has been included:

However, our findings will have immediate practical application in the development of an Australian Department of Health iSISTAQUIT social media campaign for pregnant Aboriginal and Torres Strait Islander women. These findings will guide the development of the community consultation and the media campaign to ensure that the topics explored are holistic according to the Aboriginal and Torres Strait Islander view of health and encompass the women’s requirements
